# Are Graduate Medical Trainees Prepared for the Personalized Genomic Medicine Revolution? Trainee Perspectives at One Institution

**DOI:** 10.3390/jpm13071025

**Published:** 2023-06-21

**Authors:** Elizabeth L. Kudron, Kimberly M. Deininger, Christina L. Aquilante

**Affiliations:** 1Department of Biomedical Informatics, University of Colorado School of Medicine, Aurora, CO 80045, USA; 2Colorado Center for Personalized Medicine, University of Colorado Anschutz Medical Campus, Aurora, CO 80045, USA; christina.aquilante@cuanschutz.edu; 3Department of Pharmaceutical Sciences, University of Colorado Skaggs School of Pharmacy and Pharmaceutical Sciences, Aurora, CO 80045, USA; kimberly.deininger@cuanschutz.edu

**Keywords:** personalized medicine, genetics, pharmacogenetics, medical education, residents, fellows, medical school, graduate medical training

## Abstract

Although the use of genomics to inform clinical care is increasing, clinicians feel underprepared to integrate personalized medicine (PM) into care decisions. The educational needs of physician residents and fellows, also known as graduate medical trainees (GMTs), have been overlooked. We administered an anonymous, web-based survey to all GMTs participating in training programs affiliated with our institution to evaluate their knowledge, skills, and attitudes toward PM. Of the 1190 GMTs contacted, 319 (26.8%) returned surveys. Most (88.4%) respondents reported receiving PM education in the past. Although the respondents agreed that knowledge of disease genetics (80.9%) or pharmacogenetics (87.1%) would likely lead to improved clinical outcomes, only 33.2% of the respondents felt sufficiently informed about PM. The respondents who had received PM education in residency and/or fellowship had significantly higher self-reported knowledge, ability, awareness, and adoption of PM than those who had not received this education (*p* < 0.0001, *p* < 0.0001, *p* < 0.0001, and *p* < 0.01, respectively). Targeted training is needed to improve GMTs’ confidence in interpreting and explaining genetic test results. The ideal timing for this education appears to be in residency and/or fellowship rather than in medical school.

## 1. Introduction

The era of personalized medicine has arrived with the growing availability of genomic panels and direct-to-consumer genetic testing. Personalized (or precision) medicine (PM) is an approach to clinical decision-making that accounts for interindividual variability, particularly genetic differences [1]. The increasing use of genomics to inform clinical care demands that physicians in all fields of medicine are prepared to interpret and discuss genetic testing with their patients [2,3]. As it is the role of graduate medical education (GME) to prepare the next generation of physicians for practice, educational leaders must appreciate current graduate medical trainees’ (GMTs) gaps in genomics knowledge.

Previous work has detailed practicing clinicians’ attitudes toward and knowledge of PM and genomics. Most physicians agree that genetic testing is relevant to their clinical practice [4,5,6,7] and will become an increasingly prevalent part of their clinical work [4,8]. However, both primary care and specialty providers feel underprepared to integrate genomic medicine into clinical care [4,6,7,9,10]. Practicing physicians consistently cite a lack of knowledge and low confidence in using genetics to inform clinical decision-making [2,8,10,11,12]. Studies focusing on pharmacogenetics (PGx), the use of genetics to inform medication prescribing, show that these knowledge gaps may be even larger for PGx than those described for genomics overall [4,5,6,13,14]. This can result in missed opportunities to utilize genetics in disease prevention and screening as well as medication treatment decisions [8,15,16]. Physicians recognize these deficits and feel that current medical education is not providing them with the necessary knowledge in medical genetics [4].

New and innovative educational approaches are needed to improve provider competencies in genomics [2,8,10,11,17,18,19]. This need is magnified when considering that the ongoing shortage of genetics professionals will compel non-genetics clinicians to contend with an increasing prevalence of genomics in clinical care [20,21]. Although some medical schools have responded to the call for additional training in genomics [17,22], only 21% of programs report teaching PM [23]. In terms of GME, it is unclear how programs in the United States address these topics as there are no required competencies in PM [24]. Previous research evaluating graduate medical training in genomics and PM has focused on specific programs or institutions [25,26].

Graduate medical training is a crucial time in the education of future clinicians as it provides the opportunity to assimilate knowledge obtained in the classroom into patient care [18,25,27]. Vitek et al. [6] showed that although internal medicine residents had more traditional education in PGx than attending physicians, both groups had the same levels of self-reported knowledge and skills in this domain. Thus, the integration of genetics knowledge into clinical curricula is essential to prepare the next generation of physicians to utilize this information [6,18,25,27].

Educational leaders must understand the needs and preferences of their learners to develop high-quality education that translates to improved expertise. Educational deficits in genomics knowledge have been described for medical students, pathology residents, and primary care trainees [25,26,27,28]. To date, we are unaware of any studies evaluating GMTs’ knowledge of and attitudes toward genomics, in which the investigated cohort spanned across residency programs. Here in, we describe a survey of local GMTs with the aim of assessing their perceived knowledge of genetics and PGx and their attitudes toward PM. The findings have informed the development of educational resources targeted for GMTs across specialties to promote the integration of personalized and genomic medicine into clinical practice.

## 2. Materials and Methods

### 2.1. Study Population

In this cross-sectional study, we administered a 47-item web-based survey (available in the Appendix A) to all GMTs, including interns, residents, and fellow physicians participating in training programs affiliated with the University of Colorado (CU) Anschutz Medical Campus. We received a list of GMTs enrolled in these programs from CU’s GME office.

### 2.2. Survey Instrument and Data Collection

We developed the survey instrument to assess perceived knowledge of disease genetics and PGx; attitudes towards PM; awareness of PM already being used in clinical practice; and perceived adoption of PM tools in clinical practice. We chose these focus areas with the goal of using the collected information to develop educational resources for GMTs to promote the integration of PM into clinical practice. For our survey, PM was categorized as disease genetics or PGx.

We adapted a portion of the survey from a previous study focused on the clinical utility of PGx [29]. We developed additional sections of the survey based on a thorough review of prior research evaluating medical student and clinician perspectives on PM, disease genetics, and PGx [5,13,14,28,29,30,31,32,33] as well as Everett Rogers’s diffusion of innovations (DOI) theory [34]. This theory has been used by various disciplines, including medicine, to explain how and why an innovation spreads among a specified population or system [34].

We constructed the portion of the survey based on DOI theory around two key elements that influence the adoption of an innovation: (1) attributes of innovation and (2) adopter category. Five characteristics of an innovation, referred to as attributes, contribute to its rate of adoption: (1) relative advantage—the belief that the innovation is better than the idea it supersedes; (2) compatibility—consistency of the innovation with existing values, norms, and needs of adopters; (3) complexity—the extent to which the intervention is difficult to understand and use; (4) trialability—the ability to experiment with the innovation before full adoption; and (5) observability—the degree to which benefits of the innovation are visible to potential adopters [34]. Individuals’ perceptions of these attributes have been shown to explain variability in rates of adoption [34].

Potential adopters can be classified into one of five adopter categories based on their perceived rate of adoption of an innovation [34]. These categories are: (1) innovators—those first to try an innovation; (2) early adopters—those aware of the need to change and very comfortable with adopting new ideas; (3) early majority—those who adopt new ideas before the average person but need to see evidence of success before adoption; (4) late majority—those who only adopt an innovation after the majority has tried it—skeptical of change; and (5) laggards—those who are bound by tradition and very skeptical of change [34].

One member of the study team (ELK) developed the initial survey instrument with subsequent revisions by KMD and CLA. The final survey instrument consisted of 47 questions encompassing the following domains: knowledge and confidence about PM (4 questions); perceived ability to interpret and explain a genetic test result (4 questions); awareness of PM in the clinical setting (3 questions); previous PM education and future educational preferences (3 questions); awareness and use of PM resources (8 questions); experience with genetic testing and barriers to testing (4 questions); DOI attributes of innovation and adopter category (13 questions); and demographic information (8 questions).

We administered the survey via Qualtrics (Provo, UT, USA) from December 2019 to January 2020 using a link emailed to the study population. Both the email and survey link contained information describing the purpose of the study and stating that survey completion implied consent to participate. Participation was voluntary, although the participants were incentivized with the opportunity to win 1 of 10 gift cards following survey completion. Two reminder emails were sent 7 and 35 days following the initial recruitment email. The survey closed 42 days after the initial email. The study was reviewed by the Colorado Multiple Institutional Review Board (COMIRB #19-2939) and was determined to be exempt from review. Survey responses were not linked to participant names.

### 2.3. Data Analysis

Surveys with at least 90% of the primary survey questions, excluding demographic questions, completed were included for data analysis. The external validity of the study sample was assessed by comparing the demographic characteristics of the survey respondents to the national cohort of GMTs, as reported by the Association of American Medical Colleges (AAMC, Washington, DC, USA) and the Accreditation Council for Graduate Medical Education (ACGME, Chicago, IL, USA) [35,36]. Histograms were used to visualize the heterogeneity of responses. Potential response bias could not be assessed due to the inability to obtain specific demographics for GMTs at the programs surveyed.

To simplify the data analysis, the responses were collapsed into fewer categories for selected survey questions as follows: strongly agree/agree vs. disagree/strongly disagree; age ≤ 30 years vs. age > 30 years; education in residency and/or fellowship vs. education in medical school and/or graduate school; innovators/early adopters vs. early majority vs. late majority/laggards. Response frequencies were computed for each survey question. Incomplete responses were not included in the response frequency calculations.

A knowledge summary score was calculated to assess the respondents’ confidence in their knowledge of disease genetics and PGx. Four questions asked the respondents to evaluate whether they felt informed about PM and their perceived confidence in their knowledge of the influence of genetics on disease risk, disease screening & medication therapy using a Likert scale with a range of 1–4, with 4 equating to higher confidence in their knowledge. Similarly, summary scores for perceived ability to interpret and explain a disease genetic or PGx test result; for awareness of PM in the clinical setting; and for the rate of adoption of PM were calculated using four ability, three awareness, and twelve adoption items. All scores were computed using a Likert scale with a range of 1–4 and a maximum possible score of 20.

Paired *t*-tests and Pearson chi-square tests were used to evaluate relationships between adopter category, genetic testing prevalence, barriers to genetic testing, resource utilization, and knowledge, ability, awareness, and adoption scores, with demographic variables including age, gender, previous education, and type of training. Cronbach’s alpha was computed to assess the internal consistency of knowledge, ability, awareness, and adoption items separately. A Cronbach’s alpha > 0.7 implied that the items consistently evaluated the same concept. A *p*-value < 0.05 was considered significant. All statistical analyses were performed using SAS^®^ software version 9.4 (Cary, NC, USA).

## 3. Results

### 3.1. Survey Performance

Of the 1190 residents and fellows contacted, 319 GMTs (26.8%) returned questionnaires that met criteria for inclusion in the study sample; 98.4% of respondents had fewer than 2% of missing questions. All questions demonstrated heterogeneity and performed well.

### 3.2. Respondent Characteristics

The characteristics of the respondents are shown in Table 1. Over half (59.7%) of the respondents were women, and a majority (76.2%) of respondents identified as non-Hispanic White. These are significantly higher proportions compared to the national cohort of residents, which was 45.1% female (AAMC data) and 47.5% non-Hispanic White at the time of study completion [35,36]. Almost all (98.4%) of the respondents were 40 years of age and younger. Three-fourths (75.6%) of the sample were residents or chief residents, and 72.4% were pursuing training in fields primarily treating adults. When the respondents were asked where they had previously received education or training in PM, 73.7% reported receiving education in medical or graduate school, while 50.2% reported receiving education in residency and/or fellowship (45.1% residency; 13.2% fellowship). Only 11.6% of the respondents reported not having received training in the past. Over half (60.4%) of our respondents identified as early majority when asked to classify themselves on a scale of innovativeness.

### 3.3. Knowledge, Ability, and Awareness

The statements comprising the knowledge, ability, and awareness scores are shown in Figure 1, and descriptive statistics for each computed score are shown in Table 2. Regarding confidence in their knowledge of PM, two-thirds (66.8%) of respondents did not feel sufficiently informed about this area. While two-thirds (67.4%) of respondents reported feeling confident in their knowledge of the influence of genetics on disease risk, a significantly smaller portion of respondents reported similar confidence in their knowledge of the impact of genetics on disease screening (54.9%, *p* = 0.001) and medication therapy (37.9%, *p* < 0.0001). The respondents were significantly more likely to report confidence in their ability to interpret and explain a disease genetic test result than a PGx test result (*p* < 0.0001). However, less than half (23.8–47%) of respondents reported confidence in their ability to perform any of these tasks. An even smaller number of respondents reported an awareness of PM when questioned if their patients ask about PM (24.8%), including both disease genetic testing (41.2%) and PGx testing (17.6%).

Respondents who had previously received education specific to PM in residency and/or fellowship had significantly higher knowledge, ability, and awareness scores than those who had not received education as GMTs (mean ± SD: 13.3 ± 2.6 vs. 11.7 ± 2.3; 12.8 ± 2.8 vs. 10.5± 2.8; 11.4 ± 3.1 vs. 9.6± 2.8, respectively; *p* < 0.0001 for all scores). Conversely, respondents who received PM education in medical and/or graduate school only had significantly higher ability scores compared to those who had not received education during their schooling (mean ± SD: 11.9 vs. 10.9, *p* = 0.01). Thus, PM training in medical and/or graduate school was not associated with significantly higher knowledge or awareness scores in this cohort. Respondents 31 years of age and older had significantly higher knowledge and ability scores compared to respondents who were less than 31 years old (mean ± SD: 13.0 ± 2.8 vs. 12.1 ± 2.25, *p* = 0.003 and 12.1 ± 3.1 vs. 11.2 ± 2.8, *p* = 0.009, respectively). Gender and training population (i.e., adult vs. pediatric) were not associated with higher perceived knowledge, ability, or awareness scores.

### 3.4. Adoption

The statements comprising the adoption score are shown in Figure 2, and descriptive statistics for the corresponding computed score are shown in Table 2. In terms of relative advantage, there was no significant difference in the respondents’ agreement with the benefits of disease genetic vs. PGx testing likely outweighing the risks or costs. However, significantly more respondents agreed that there is sufficient evidence to support ordering disease genetic tests compared to PGx tests (*p* < 0.00001). For compatibility, two-thirds (65.0%) of respondents felt that their clinical program would be able to integrate PM into its workflow. Regarding complexity, there was no significant difference in respondents’ agreement that they would still perform a disease genetic vs. a PGx test even if the results were difficult to interpret. However, only half of the respondents would still perform a genetic test in this scenario (53.8% for disease genetic test and 47.3% for PGx test). Most respondents (80.9–91.8%) agreed with the four observability statements, which anchored on the idea that PM and knowledge of a patient’s genetics would change clinical management and improve clinical outcomes.

Respondents who had previously received education specific to PM in residency and/or fellowship had significantly higher adoption scores than those who had not received this training (mean ± SD: 14.1 ± 2.0 vs. 13.5 ± 2.2, *p* < 0.01). However, there was no significant difference in adoption scores for respondents who received training in medical and/or graduate school vs. those who did not. Gender, training population, and age were not significantly associated with adoption scores.

There was a significant difference in adoption scores based on adopter category. Respondents who identified as innovators/early adopters had significantly higher adoption scores than those who identified as late majority/laggards (mean ± SD: 14.6 ± 2.5 vs. 13.0 ± 1.9, *p* < 0.0001). However, adopter category was not significantly associated with knowledge, ability, or awareness scores. Previous education, gender, age, and training population were not associated with adopter category.

### 3.5. Genetic Testing: Prevalence and Barriers

Approximately half of the respondents (47.0%) reported ordering genetic testing in the 12 months prior to completing our survey. Respondents who identified as female, aged 31–60, and enrolled in pediatric training programs were significantly more likely than those who identified as male, aged less than 31, and in adult or combined training programs to have ordered genetic testing in the past year (52.2% vs. 40.3%, *p* = 0.04; 55.7% vs. 40.7%, *p* = 0.008; 70.8% vs. 40.9%, *p* < 0.0001, respectively). Additionally, respondents who reported receiving PM education in residency and/or fellowship were significantly more likely to have ordered genetic testing than those who had not received similar education (62.6% vs. 33.3%, *p* < 0.0001). PM education received in medical and/or graduate school was not significantly associated with ordering a genetic test in the past 12 months. All computed scores were significantly higher among those who reported test ordering in the past year compared to those who did not report test ordering: (mean ± SD) knowledge 13.2 ± 2.7 vs. 11.9 ±2.3, *p* < 0.0001; ability 12.3 ± 3.2 vs. 11.0 ± 2.7, *p* < 0.0001; awareness 11.1 ±3.0 vs. 9.9 ± 3.1, *p* = 0.0005; and adoption 14.0 ± 1.9 vs. 13.6 ± 2.3, *p* = 0.03.

Perceived barriers to genetic testing are shown in Figure 3. More than half of the cohort responded that the top three barriers to genetic testing were (1) uncertainty about the clinical value of the test (55.8% for disease genetic and 64.9% for PGx testing); (2) the cost to the patient (58.6% for disease genetic and 51.1% for PGx testing); (3) and not knowing what test to order (51.7% for disease genetic and 53.6% for PGx testing). More than 1/3 of the cohort (36.1% for disease genetic and 43.9% for PGx testing) responded that not knowing how to interpret test results was a barrier to testing. The respondents were significantly more likely to identify uncertainty about the test’s clinical value and not knowing how to interpret test results as barriers to PGx testing than to disease genetic testing (*p* = 0.02 and *p* = 0.04, respectively).

Respondents who had received PM education in residency and/or fellowship were significantly less likely to report a lack of knowledge about test ordering (*p* = 0.008) and interpretation (*p* = 0.02) as barriers to genetic testing compared to those who had not received education in residency and/or fellowship. This association was not observed for respondents who reported receiving education in graduate and/or medical school. Correspondingly, respondents who reported a lack of knowledge of test ordering and interpretation as barriers had significantly lower knowledge (mean ± SD; 11.9 ± 2.2 vs. 13.5 ± 2.8 and 11.6 ± 2.2 vs. 13.3 ± 2.6, respectively; *p* < 0.0001) and ability (10.9 ± 2.6 vs. 12.8 ± 3.2 and 10.4 ± 2.3 vs. 12.8 ± 3.1, respectively; *p* < 0.0001) scores than those who did not report these concerns. Respondents who identified ethical concerns as a barrier to testing had significantly higher knowledge (mean ± SD; 14.4 ± 2.0 vs. 12.4 ± 2.6; *p* = 0.0025) and awareness (13.0 ± 3.9 vs. 10.3 ± 3.0; *p* = 0.0006) scores than those who did not identify this barrier.

### 3.6. Resource Utilization

We asked the respondents about their awareness and use of widely known genetic testing and PGx resources (Table 3). The most used resources were consultation with experts (42.5%) and the scientific literature (32.4%). Compared to those who had not received PM education in residency and/or fellowship, respondents who had received this education were significantly more likely to have been aware of and/or used the following resources: ClinGen (*p* = 0.01), ClinVar (*p* = 0.02), FDA drug label (*p* = 0.02), and the scientific literature (*p* < 0.0001). Similar associations were not observed for respondents who reported receiving education in graduate and/or medical school.

Close to half (40%) of the respondents reported using no resources for information about PGx in the past. Those who had not received PM education in residency and/or fellowship were significantly more likely to report using no resources than those who had received this education (*p* < 0.0001). Similarly, those who had not received education in graduate and/or medical school were significantly more likely to report using no resources than those who had received this education (*p* = 0.004).

### 3.7. Educational Preferences

We asked the respondents how they would prefer to receive future PM education and how long they would be willing to spend learning about this topic. The respondents’ most preferred formats for receiving education were in-person lectures (62.6%), brief online modules (52.5%), and written materials (43.1%). Half (47.8%) of the respondents were willing to spend 1–2 h learning about PM, while one-third (32.7%) were only willing to spend less than one hour on the topic.

## 4. Discussion

This is the first cross-sectional survey to explore GMTs’ knowledge of and attitudes toward genomics and PM, in which the respondents spanned across residency programs. It is essential to understand the educational needs of GMTs for the next generation of physicians to be prepared to integrate genomic medicine into clinical care. We found that most respondents had received PM education or training in the past. However, two-thirds of the respondents did not feel sufficiently informed about PM, and less than half of the respondents reported confidence in their ability to interpret or explain a disease genetic or PGx test result. GMTs who had received PM education in residency and/or fellowship had significantly higher self-reported knowledge, ability, awareness, and adoption of PM than those who had not received this education. A similar trend was not observed among GMTs who had vs. had not received education in medical and/or graduate school. The findings from our study demonstrate that education delivered in graduate medical training is critical to improving the utilization of personalized and genomic medicine in clinical care.

It is well established that practicing physicians have low levels of knowledge related to genetic testing and feel underprepared to integrate PM into clinical care [4,6,7,8,9,10,11]. However, it is unclear if educational reforms emerging from this literature have impacted the future generation of physicians as GMTs’ needs have generally not been explored. In 2021, Haspel et al. found that first-year pathology residents had significantly lower knowledge of genetics than other topics [26]. Our study expands upon this observation by exploring GMTs’ knowledge of and attitudes toward genomics across residency programs and years in training. We found that two-thirds (66.8%) of the respondents did not feel sufficiently informed about PM. Less than half of the respondents felt confident in interpreting and explaining a disease genetic or PGx test. GMTs enrolled in adult and pediatric programs reported similar confidence levels in these tasks. An ongoing series evaluating genetic testing cases demonstrated that a lack of knowledge of genomics can lead to inappropriate testing, incorrect diagnoses, and delays in disease treatment [15,16]. Thus, GME educational interventions designed to improve PM knowledge gaps across disciplines could potentially have significant implications for patient care.

To create effective educational reforms, it is essential to know where the most significant knowledge gaps exist. While studies of physician preparedness for PM have focused on genomics as a whole [8,10,11], we separated this domain of medicine into disease genetics and PGx. Dividing these segments of genomic medicine allowed us to identify GMTs’ educational gaps and needs more specifically. We found that most GMTs felt confident in their knowledge of the influence of genetics on disease risk. The respondents were significantly more likely to report confidence in their ability to interpret and explain a disease genetic test result than a PGx test result. Similarly, the respondents were significantly more likely to agree that there is sufficient evidence to order a disease genetic test than a PGx test. These findings are consistent with previous research showing that physicians generally report lower knowledge of PGx compared to other aspects of genomics [4,5,6,13,37].

Given these gaps in genomics knowledge, it is not surprising that there has been a call for reforms to improve genetics education [4,11,17,18,19,26,38]. However, it is unclear how and when this should occur. Our study provides unique insight into the ideal timing of this education. We found that those who received PM training in residency and/or fellowship reported significantly higher knowledge, ability, awareness, and adoption of PM than those who had not. We did not observe similar trends among those who had vs. had not received PM education in medical and/or graduate school. This may be explained by the fact that training in genomics and PM varies across US medical schools [18,23,38]. A 2015 study found that only 21% of medical schools provided PM education [23]. The training that is offered is mainly in the pre-clinical years and lacks focus on clinical application [23,28,39].

Conversely, graduate medical training inherently targets the provision of clinically-relevant education [40]. This likely accounts for why our survey respondents who had received education during this key phase were significantly less likely to report a lack of knowledge of test ordering or interpretation as barriers to genetic testing than those who had not received this education. Thus, our findings support previous research showing that clinically integrated training increases confidence in using genomics [21,25,41] and highlight the importance of incorporating genomics education into clinical applications. This may fit most naturally into graduate medical training, given how education is delivered during this time. Since GME varies widely across programs and institutions, leaders looking to introduce or enhance curricula in genomics and PM would benefit from considering the specific educational gaps of their GMTs.

To help educators reform genomics curricula, we explored GMTs’ preferred educational formats and time spent learning about this topic. The respondents’ most preferred formats for receiving education were in-person lectures and short online modules. These findings are consistent with previous research evaluating practicing physicians’ preferences for genomics education delivery [8,11,42]. It is not clear from the literature how long GMTs or practicing physicians would be willing to spend learning about this topic. We found that most respondents would be willing to spend no more than two hours studying PM and genomics. The short amount of time our respondents would consider spending on this topic highlights the need for education to demonstrate the importance of learning about this domain of medicine as its prevalence grows in a variety of clinical specialties. Conversely, it also emphasizes that GMTs have time constraints, and that genomics education may have to be creatively threaded throughout other scheduled activities, e.g., a short pharmacogenomic vignette during didactic cardiology education.

A strength of our study was utilizing the DOI theory attributes of innovation to evaluate the adoption of PM. DOI theory purports that opinions on each attribute of innovation can promote or alienate an innovation’s uptake [34]. In our cohort, views about the observability of PM suggest that GMTs recognize the significant influence that genomics may play in clinical care. However, our survey results indicate that integrated educational resources and clinical decision support tools are needed to reduce the complexity of genomic testing interpretation and to highlight the relative advantages of genomic testing to inform clinical care decisions. These findings support previous studies that have emphasized the importance of integrating education into clinical workflows to promote the use of PM in clinical care [2,8,10,11,43]. It is important to consider that our respondents perceived higher complexity and lower relative advantage of PGx testing than disease genetic testing. Thus, there is an even greater need for interventions to promote PGx testing when appropriate.

It is crucial to consider the results of our study in the context of certain limitations. Our study used a convenience sample, and the results describe the views of GMTs affiliated with one large university system. Our institution is an early adopter of PM with a team focused on integrating genomics into clinical care and delivering the education needed to do so. Thus, our study cohort may have higher self-reported knowledge, ability, awareness, and adoption of PM than GMTs in other programs. Our survey should be repeated in different settings to evaluate whether our findings remain consistent across GMT programs. Furthermore, 73.2% of eligible participants did not respond to our survey. GMTs more interested in PM and genomics may have been more likely to complete our survey. This could have resulted in overestimating GMTs’ knowledge of and positive attitudes toward PM.

Our study was designed to assess self-reported knowledge and ability. It is common to use self-reported confidence when considering physician preparedness for PM. However, self-reported knowledge may not correlate with actual genomics knowledge and skills. In the future, case scenarios and clinically relevant knowledge questions could be added to our survey to assess actual knowledge and skills pertinent to genomic medicine.

Recency bias may have also impacted our results. GMTs who recently received training in PM may have been more likely to report higher knowledge, ability, and awareness of PM. Future studies would benefit from comparing similar cohorts in various stages of medical training to ascertain the best time for integration of PM education.

In summary, our study shows that although most GMTs have received PM education, this training has not adequately prepared them to utilize genomics in the clinical setting. GMTs agree that knowledge of genetics leads to improved clinical outcomes. However, they do not feel sufficiently informed about PM. These educational gaps may be improved through targeted, thoughtfully timed training that focuses on the clinical applicability of genomic medicine. GMTs who received PM education during their graduate medical training are significantly more likely to report higher knowledge, ability, awareness, and adoption of PM than those who have not received this training. A similar finding was not observed for those who had vs. had not received education in medical school. Thus, genomics education delivered in residency and/or fellowship has the potential to develop clinicians who are confident in their abilities to integrate genomic medicine into clinical care. Simply focusing on educational reform to improve genomics training in medical school may result in missed opportunities to best prepare the future generation of physicians for an era of PM.

## Figures and Tables

**Figure 1 jpm-13-01025-f001:**
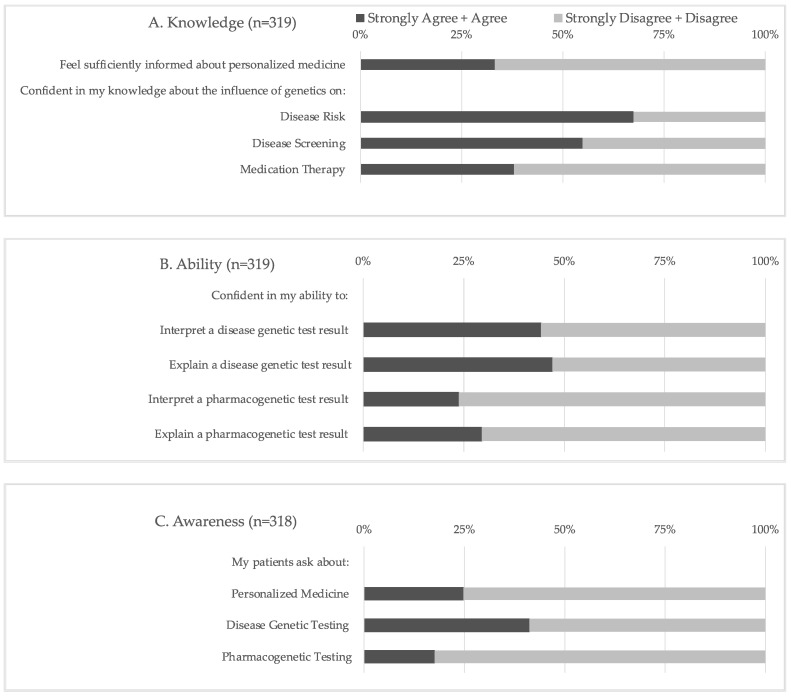
Knowledge, ability, and awareness assessments. Four statements comprise the (**A**) knowledge and (**B**) ability scores; three statements comprise the (**C**) awareness score.

**Figure 2 jpm-13-01025-f002:**
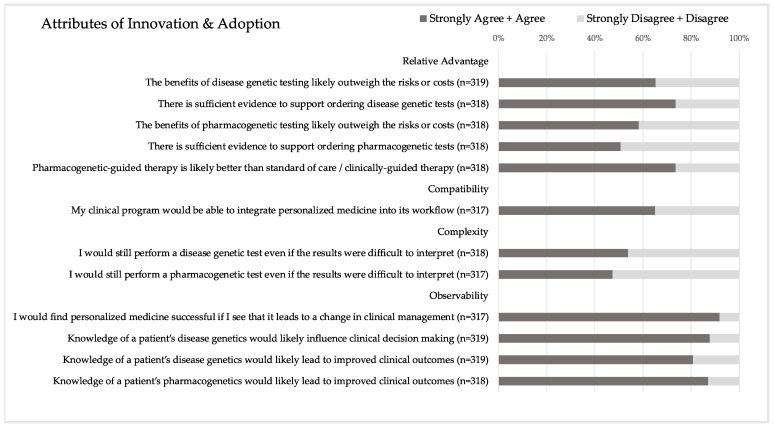
Attributes of innovation and adoption assessment. Twelve statements evaluate the attributes of innovation and comprise the adoption score.

**Figure 3 jpm-13-01025-f003:**
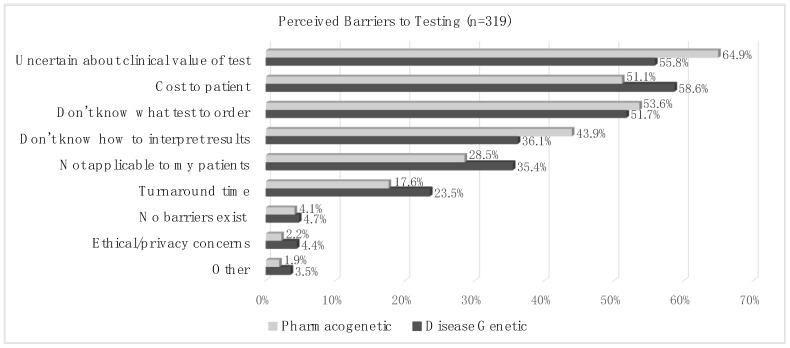
Perceived barriers to genetic testing. Percentage of respondents who identified barriers to pharmacogenetic or disease genetic testing.

**Table 1 jpm-13-01025-t001:** Characteristics of respondent population.

Characteristic (*n*)	N (%)
Gender (*n* = 313)	
Male	126 (40.3%)
Female	187 (59.7%)
Age (Years) (*n* = 317)	
21–30	176 (55.5%)
31–60	141 (44.5%)
Race (*n* = 319)	
White	243 (76.2%)
Asian	53 (16.6%)
Black, Native American, or Pacific Islander	5 (1.6%)
Other	8 (2.5%)
Prefer not to answer	20 (6.3%)
Ethnicity (*n* = 297)	
Hispanic	13 (4.4%)
Non-Hispanic	268 (90.2%)
Prefer not to answer	16 (5.4%)
Clinical Role (*n* = 319)	
Resident	213 (66.8%)
Chief Resident	28 (8.8%)
Fellow	78 (24.4%)
Post-Graduate Level (*n* = 318)	
PGY1	66 (20.8%)
PGY2	68 (21.4%)
PGY3	63 (19.8%)
PGY4	43 (13.5%)
PGY5	42 (13.2%)
PGY6	20 (6.3%)
PGY7	9 (2.8%)
PGY8, 9, and 10	7 (2.2%)
Training Population (*n* = 319)	
Adult	231 (72.4%)
Pediatric	73 (22.9%)
Adult and Pediatric	15 (4.7%)
Previous Personalized Medicine Education or Training (*n* = 319)	
Medical or Graduate School	235 (73.7%)
Residency and/or Fellowship	158 (49.5%)
Grand Rounds or Lectures	61 (19.1%)
Conferences	44 (13.8%)
Undergraduate School	42 (13.2%)
Institutional Initiatives	12 (3.8%)
CME Programs	9 (2.8%)
No Prior Education	37 (11.6%)
Other ^1^	14 (4.4%)
Adopter Category (*n* = 318)	
Innovators	5 (1.6%)
Early Adopters	68 (21.4%)
Early Majority	192 (60.4%)
Late Majority	50 (15.7%)
Laggards	3 (0.9%)

^1^ Other: Previous research or employment.

**Table 2 jpm-13-01025-t002:** Knowledge, ability, awareness, and adoption scores.

Summary Score	Number of Items	n	Mean	SD ^1^	Range	Cronbach’s Alpha
Knowledge	4	319	12.49	2.56	5–20	0.74
Ability	4	319	11.63	2.99	5–20	0.84
Awareness	3	318	10.45	3.11	5–20	0.75
Adoption	12	319	13.78	2.13	5–20	0.88

^1^ Abbreviation. SD: standard deviation.

**Table 3 jpm-13-01025-t003:** Awareness and utilization of genetic testing and pharmacogenomic resources.

Resource	Utilized in Practice n (%)	Aware of Resource n (%)	Not Aware of Resource n (%)
Clinical Pharmacogenetics Implementation Consortium (CPIC)	3 (0.9%)	15 (4.7%)	300 (94.3%)
Pharmacogenomics Knowledgebase (PharmGKB)	3 (0.9%)	14 (4.4%)	301 (94.7%)
Dutch Pharmacogenetics Working Group (DPWG)	3 (0.9%)	9 (2.8%)	306 (96.2%)
ClinGen	17 (5.3%)	74 (23.2%)	228 (71.5%)
ClinVar	15 (4.7%)	35 (11.0%)	268 (84.3%)
American College of Medical Genetics (ACMG)	12 (3.8%)	68 (21.3%)	239 (74.9%)
Implementing Genomics in Practice (IGNITE)	3 (0.9%)	17 (5.4%)	298 (93.7%)
Clinical Decision Support in EHR ^1^	31 (9.8%)		
Consultation with Experts or Peer Discussions	135 (42.5%)		
National Guidelines	60 (18.9%)		
FDA Drug Labels	17 (5.4%)		
Scientific Literature	103 (32.4%)		
Other Resources	30 (9.4%)		
No Resources	128 (40.3%)		

^1^ Abbreviation. EHR: electronic health record.

## Data Availability

This work does not include any clinical data. The data are available for review upon request; inquiries can be directed to the corresponding author.

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
