# Peer review of "Are Graduate Medical Trainees Prepared for the Personalized Genomic Medicine Revolution? Trainee Perspectives at One Institution"

_jpm, 2023, doi:10.3390/jpm13071025_

Round 1
Reviewer 1 Report
This manuscript presents the results of a survey to assess the knowledge of genetic medicine trainees (GMTs) regarding genetics and personalized medicine. The authors emphasize on the significance of educational needs of GMTs for the effective incorporation of personalized medicine into clinical practice. Overall, this study provides valuable understanding into GMT knowledge toward genomics and personalized medicine. By emphasizing the importance of education during postgraduate medical education, this study contributes to a broader understanding of personalized medicine into clinical practice. Taking these comments into account increases the quality and impact of the manuscript.
It would be helpful to deliver more evidence on the specific aspects of personalized medication covered by the training that GMTs. This would deliver a better knowledge of gaps and areas that need further stress.
The study focuses on education received through residency and fellowship, but does not study the potential impact of personalized medicine during medical or graduate school.
This feature should be discussed further, especially if there were limitations in assessing the impact of the education received at various stages of the training. How might the inadequate knowledge of GMT affect the delivery of personalized treatment and patient outcomes? Addressing these points would enhance the implication and significance of the study.
Well written.
Reviewer 2 Report
- Several grammatical errors that make it difficult to understand.
- It is better for the authors to revise and expand their conclusion
- New references for 2022, 2023 will be added to text
- Future aspects are missing
- The first time you include acronyms within the text, you have to write them in full. After that, you should report them as abbreviations only.
In the introduction, in the last paragraph, the importance of the issue and the necessity of implementation should be explained. Also explain why you did the present study
in the work method; Was it random in collecting the samples? Sample size formula
The discussion needs to be fundamentally rewritten:
The results of other studies, whether similar or different, should be brought and compared and inferred, and what is the difference between the current article and those articles?
Mention each of your important findings, then compare with others and mention your interpretation and inference of the results and causes of similarities and differences.
Indicate the application of the results
The limitations of the study should be mentioned
Running title should be added
The type of study should be mentioned.
It should also be mentioned the sampling method and where and what center the patients were selected from.
On what basis and how was the sample size selected? The number of samples should be justified in a scientific way. Please, according to the main purpose of the study, add the appropriate formula and valid parameters by mentioning the source.
Wishing you all of best.
- Several grammatical errors that make it difficult to understand, please check.
- It is better for the authors to revise and expand their conclusion
- New references for 2022, 2023 will be added to text
- Future aspects are missing
- The first time you include acronyms within the text, you have to write them in full. After that, you should report them as abbreviations only.
In the introduction, in the last paragraph, the importance of the issue and the necessity of implementation should be explained. Also explain why you did the present study
in the work method; Was it random in collecting the samples? Sample size formula
The discussion needs to be fundamentally rewritten:
The results of other studies, whether similar or different, should be brought and compared and inferred, and what is the difference between the current article and those articles?
Mention each of your important findings, then compare with others and mention your interpretation and inference of the results and causes of similarities and differences.
Indicate the application of the results
The limitations of the study should be mentioned
Running title should be added
The type of study should be mentioned.
It should also be mentioned the sampling method and where and what center the patients were selected from.
On what basis and how was the sample size selected? The number of samples should be justified in a scientific way. Please, according to the main purpose of the study, add the appropriate formula and valid parameters by mentioning the source.
